# DNA copy number changes define spatial patterns of heterogeneity in colorectal cancer

Soulafa Mamlouk[1,2,3,*], Liam Harold Childs[4,*], Daniela Aust[2,5,6,*], Daniel Heim[1], Friederike Melching[1], Cristiano Oliveira[7], Thomas Wolf[2,7], Pawel Durek[8], Dirk Schumacher[1,2,3], Hendrik Bläker[1,2], Moritz von Winterfeld[1,†], Bastian Gastl[1,9], Kerstin Möhr[1], Andrea Menne[1,2,3], Silke Zeugner[5], Torben Redmer[1,2,3], Dido Lenze[1], Sascha Tierling[10], Markus Möbs[1], Wilko Weichert[2,11], Gunnar Folprecht[12], Eric Blanc[13,14], Dieter Beule[13,15], Reinhold Schäfer[1,2,3], Markus Morkel[1], Frederick Klauschen[1], Ulf Leser[4] & Christine Sers[1,2]

Genetic heterogeneity between and within tumours is a major factor determining cancer progression and therapy response. Here we examined DNA sequence and DNA copy-number heterogeneity in colorectal cancer (CRC) by targeted high-depth sequencing of 100 most frequently altered genes. In 97 samples, with primary tumours and matched metastases from 27 patients, we observe inter-tumour concordance for coding mutations; in contrast, gene copy numbers are highly discordant between primary tumours and metastases as validated by fluorescent *in situ* hybridization. To further investigate intra-tumour heterogeneity, we dissected a single tumour into 68 spatially defined samples and sequenced them separately. We identify evenly distributed coding mutations in *APC* and *TP53* in all tumour areas, yet highly variable gene copy numbers in numerous genes. 3D morpho-molecular reconstruction reveals two clusters with divergent copy number aberrations along the proximal–distal axis indicating that DNA copy number variations are a major source of tumour heterogeneity in CRC.

[1] Institute of Pathology, Charité Universitätsmedizin Berlin, Berlin 10117, Germany. [2] German Cancer Consortium (DKTK), Heidelberg 69120, Germany. [3] German Cancer Research Center (DKFZ), Heidelberg 69120, Germany. [4] Knowledge Management in Bioinformatics, Humboldt University of Berlin, Berlin 10099, Germany. [5] Institute for Pathology, University Hospital Carl Gustav Carus, Technische Universität Dresden, Dresden 01307, Germany. [6] NCT Biobank Dresden, University Hospital Carl Gustav Carus, Technische Universität Dresden, Dresden 01307, Germany. [7] Institute of Pathology, University of Heidelberg, Heidelberg 69120, Germany. [8] Experimental Rheumatology, German Rheumatism Research Centre, Berlin 10117, Germany. [9] BSIO Berlin School of Integrative Oncology, University Medicine Charité, Berlin 13353, Germany. [10] Department of Genetics/Epigenetics, FR8.3 Life Sciences, Saarland University, Saarbrücken 66123, Germany. [11] Institute of Pathology, Technical University Munich, Munich 81675, Germany. [12] University Hospital Carl Gustav Carus, University Cancer Center/Medical Dpt. I, Dresden 01307, Germany. [13] Core Unit Bioinformatics, Berlin Institute of Health, Berlin 10117, Germany. [14] Charité Universitätsmedizin Berlin, Berlin 10117, Germany. [15] Max-Delbrück-Center for Molecular Medicine, Berlin 13125, Germany. * These authors contributed equally to this work. † Present address: Institute of Pathology, Uniklinik Köln, Köln 50937, Germany. Correspondence and requests for materials should be addressed to C.S. (email: christine.sers@charite.de).

Cancer heterogeneity is a major driving force for tumour progression, metastasis and therapy resistance, and poses a major challenge in personalized cancer medicine[1]. Individual patterns of single-nucleotide variations (SNV) and DNA copy number variations (CNV) occur in tumours both between patients (inter-patient) and within single patients (inter-tumour). In addition, spatially separated, molecularly diverse areas may exist within tumours, defined as intra-tumour heterogeneity[2]. These levels of heterogeneity are the result of intrinsic mechanistic processes, such as inherent genomic instability, clonal selection and competition, and tissue specific tumour-host interactions[3–7].

Colorectal cancer (CRC) is one of the best understood solid cancers[8,9]. The mutational status of CRC determines targeted treatment options. RAS and BRAF wild type tumours frequently respond to anti-EGFR therapy, whereas mutant tumours are refractory due to primary resistance. Importantly, the emergence of mutations within the KRAS, NRAS and BRAF oncogenes[10–12] in tumours previously tested to be RAS and BRAF wild type often results in secondary resistance. However, high-depth massive parallel sequencing analyses revealed undetected low-frequency KRAS mutant clones already present in primary cancers[10,11]. This suggests the existence of intra-tumour heterogeneity and subsequent clonal selection for these mutations as a major determinant of therapy outcome in CRC. Importantly, spatially different areas of a tumour may harbour individual mutational patterns, enabling outgrowth of sub-clones, which slip through mono-sampling diagnostics[3,13].

Yet, the mutational status of KRAS, NRAS, BRAF and other proto-oncogenes and tumour suppressor genes does not account for all cases of resistance to targeted therapy. A particularly important mechanism besides genomic variations within coding regions are variations in gene copy numbers (CNVs). These comprise focal amplifications, aneuploidy or loss of heterozygosity, for instance in tumour suppressors like APC and TP53. The impact of CNVs on therapy resistance has already been proven; for instance, CRC patients with amplifications in KRAS, ERBB2, MET and FGFR1 show poor prognosis and resistance to anti-EGFR therapy[14,15].

Thus, genetic heterogeneity at both the SNV and the CNV level can be a major obstacle to successful therapy and is a challenge for mono-sampling diagnostics. Although several studies have looked at genetic heterogeneity in CRC[16–18], open questions remain concerning the number of genes involved, the distribution and nature of genetic aberrations between primary and metastatic lesions, and sampling effects. Furthermore, information on molecular differences between primary, synchronous and metachronous tumour samples following therapy is still limited.

To address the contributions of SNVs and CNVs to CRC heterogeneity, we performed a detailed analysis of inter-patient, inter-tumour and intra-tumour heterogeneity. Using a CRC-specific DNA sequencing panel covering 100 genes, we assessed mutational patterns and CNVs using ultra-deep sequencing and validated CNVs by fluorescent in situ hybridization (FISH). We examined 97 samples from 27 CRC patients to study inter-patient and inter-tumour heterogeneity (Fig. 1a,b) where we found little genetic heterogeneity at the mutation level but strong CNV heterogeneity between and within each patient. Furthermore, we performed an in-depth analysis of 68 samples from a single primary tumour to assess intra-tumour heterogeneity. Serial sections of the tumour, each divided into the luminal, deep invasive and lateral fronts, were sequenced along the proximal–distal axis of the colon (Fig. 1c–e) and genetic alterations, validated by FISH and shallow whole-genome sequencing (WGS), were mapped to a three-dimensional (3D)-model of the tumour. We find a homogeneous distribution of genetic mutations, yet strongly diverging numbers of DNA copy numbers in multiple genes implicated in tumour progression and therapy response, such as CDX2, CARD11, MMP9 and BRCA2. This work indicates that regional differences in gene copy numbers are an important aspect of tumour heterogeneity in CRC. Our results thus propose implementation of broader clinical routines taking into account both DNA mutations and copy number changes.

## Results

**SNVs are highly concordant between tumours of the same patient.** To evaluate inter-patient and inter-tumour heterogeneity, we analysed 27 CRC patients (40% stage III; 60% stage IV), each with a primary tumour and at least one metastasis to the lymph node, liver, retro peritoneum, lung, skin, small bowel, soft tissue, ovary, uterus, brain or rectum together with matched healthy tissue. (Fig. 1a,b and Supplementary Fig. 1).

We performed targeted high-depth sequencing (mean depth 1,500 reads) for genomic areas comprising 100 genes using a custom-designed CRC panel (Supplementary Fig. 2a and Supplementary Data 1). We detected 88 distinct non-synonymous mutations in 20 genes when compared with the respective normal tissue samples (Fig. 2a). TP53 was the most commonly mutated gene, aberrant in 21 patients (78% of cohort), followed by APC in 19 patients (70%). KRAS (12 patients, 44%) harboured the commonly described variants G12D, G12C, G12V, G13D, Q61H, K117N and A146V. FBXW7 was mutated in six patients (22%), and four of these exhibited the known S582L variant. In addition to these commonly detected CRC mutations[17,19], we identified further mutations, for instance in CDX2, WFDC2 and MMP17. The CDX2 mutation results in an amino acid deletion (p.248delK) located in the homeobox domain of the encoded protein. One patient carries a WFDC2 (also known as HE4) mutation (c.C13A, p.R5S). A similar mutation (R5H) was described previously in the large intestinal cell line Gp5D (ref. 20).

Only four patients (15%) displayed inter-tumour heterogeneity of SNVs, defined as discordant mutational patterns in the samples originating from one patient (Fig. 2b). Patient D3 carries a private mutation in MMP17 in the primary tumour (c.G371A, p.R124K), which is absent from the synchronous metastasis. Three patients carry mutations in metastatic samples not detected in the primary tumour samples: TCF7L2 (c.T631G, p.F211V, liver metastasis of patient D23), GNAS (c.G2531A, p.R844H in one of three lymph node metastases of patient D26), CARD11 (c.C1267T, p.R423W) and TP53 (c.A736G, p.M246V; Fig. 2b). The latter two mutations were found in a post-chemotherapy lung metastasis of patient D27, but were absent in the matched pre-chemotherapy primary, lymph node and brain metastasis samples. All discordant mutations were verified by either Sanger sequencing, primer extension HPLC-based SNuPE (Supplementary Fig. 3) or by re-sequencing on a MiSeq Illumina platform.

In summary, our deep sequencing results indicate mainly concordant SNV patterns within each CRC patient, even in metachronous metastases resected long after the primary tumour. For example, patient D41 displayed concordant SNV patterns in a small bowel metastasis removed 60.5 months after the primary colon tumour and had received both targeted and chemotherapy during that period. We also examined the cohort for silent mutations as well as variants residing in intron regions covered by the panel (Supplementary Fig. 4). While primary and metastatic samples of each CRC patient were highly concordant with respect to driver-gene alterations, they display clear differences in intronic and synonymous mutations.

**CNVs are highly discordant between tumours from the same patients.** Next we determined CNVs from our panel-sequencing

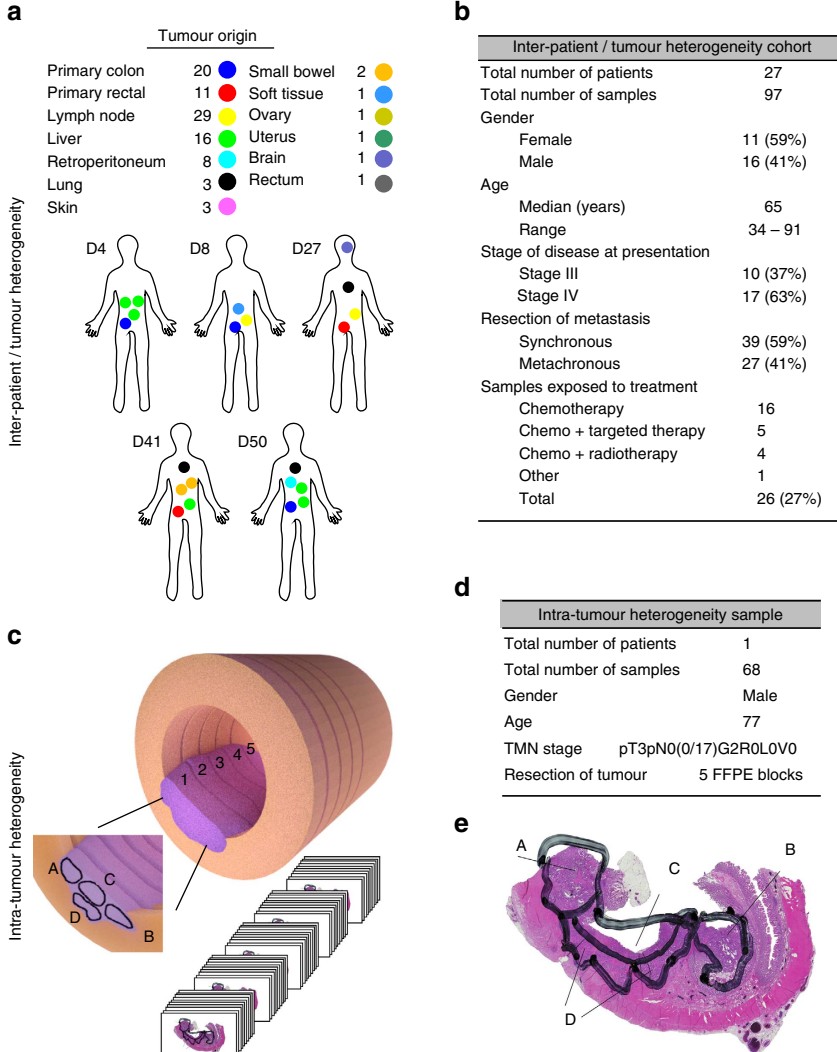

**Figure 1 | Experimental set up for the detection of tumour heterogeneity in CRC.** (**a**) Inter-patient and inter-tumour heterogeneity was investigated in 27 patients with 31 primary tumours and 66 metastases from different organs. Five patients are shown in detail (D4, D8, D27, D41 and D50). (**b**) Detailed information on the cohort studied. (**c**) Intra-tumour heterogeneity was investigated in a single stage II CRC tumour (tumour depicted in pink, colon in orange) which was isolated entirely and separated into 5 individual blocks. Each block was sectioned completely and each set of 15 sections was grouped together. Each section was divided into four compartments (A, B, C and D), resulting in a total of 68 samples used for massive parallel sequencing analysis. (**d**) Detailed information on the stage II CRC tumour studied. (**e**) H&E staining of one representative section depicting compartments A, B, C and D corresponding to the left and right lateral tumour regions, and the luminal and deep invasive front, respectively.

data by employing CNVPanelizer[21], an algorithm specifically designed to estimate copy number frequencies using targeted massive parallel sequencing data. We found CNVs deviating from the diploid state of normal tissues in 74 of 100 genes represented by the CRC panel. The most frequent gene copy gain was detected for *CDX2* (13q12.3; 82% of samples) and *WFDC2* (20q13.12; 56%). The tumour suppressor *SMAD4* located on chromosome 18q21.1 was frequently affected by gene copy loss (53%; Fig. 2c). Furthermore, we found recurring amplifications of chromosomal regions carrying *AMER1, CARD11, PTK2, PREX2* and *EGFR*.

We found a high level of inter-patient CNV heterogeneity, but in contrast to the SNV data presented above, we also detected a high level of inter-tumour CNV discordance within individual patients (Fig. 3, Supplementary Fig. 5). For instance, patient D4 displays multiple DNA copy number differences between the primary tumour (D4P7) and the three metastases (D4M8, D4M9 and D4M10). *CARD11* CNV was diploid in the primary tumour, but increased in all three metastases. *SMAD4* copy numbers decreased specifically in the metastasis D4M10. *MMP9* showed a

copy number increase in metastasis D4M9 and D4M10, but not in the primary and D4M8. Note that D4M8 was resected 2 months before the primary carcinoma (D4P7), while D4M9 and D4M10 were removed 5 months and 2 years, respectively, after the primary carcinoma and intermittent FOLFOX therapy. *CARD11* showed a copy number gain in the retroperitoneal metastasis of patient 34 (D34M107), but not in the primary tumour (D34P105). Notably, we did not detect any specific CNV pattern typical of either metastases or primary tumours (see 'Methods' section 'study of genetic and patient characteristics').

To validate our findings regarding CNVs identified by CNVpanelizer, we investigated *SOX11, CDX2, MMP9, CARD11* and *EDEM2* copy numbers in 41 tumour sections by gene-specific FISH. *SOX11* exhibited two alleles both by CNVPanelizer and FISH. In agreement with CNVPanelizer data, we observed amplification in *CDX2* and *MMP9* in all samples from patients D8 and D2, respectively. FISH analysis also confirmed the high degree of inter-tumour CNV (Fig. 4a,b). For instance, we found the metastasis-specific copy number gain for *CARD11* in patient

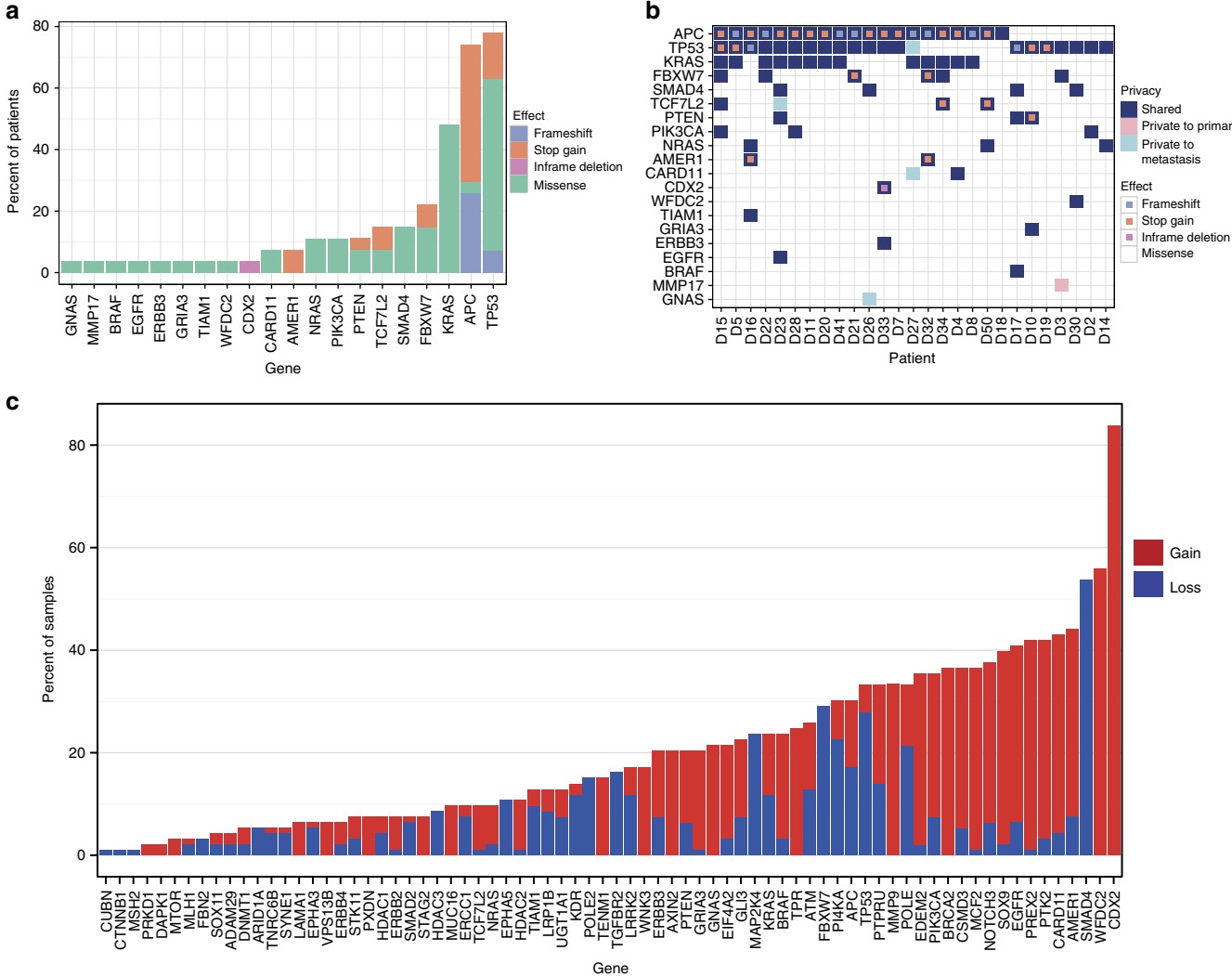

**Figure 2 | Overview of single nucleotide (SNV) and gene copy number variant (CNV) distributions in the inter-tumour heterogeneity study.**
(**a**) Frequencies and distribution of genomic variants across all patients: frame-shift (purple), stop-gain (orange), deletion (pink) and missense mutation (green). (**b**) Patterns of concordance and discordance within each individual patient. Colour code: shared mutations, that is, found in all samples from one patient (dark blue); private mutation, that is, restricted to the primary tumour (pink); private mutation to metastasis (light blue). (**c**) Distribution of CNV, called by CNVPanelizer, across all genes within the patient cohort. Colour code: gain of gene copy number (red); loss of gene copy number (blue).

D4 for both liver metastases (average copy number 3.26, 3.78 and 4.54, respectively). Furthermore, we detected CNV gains and inter-tumour CNV heterogeneity in multiple patients by FISH, which did not reach significance in the CNVPanelizer analysis, such as different levels of amplification of *CDX2* (patient D8, see above) and *EDEM2* in patient D10. Overall, FISH and CNVPanelizer data exhibited good correlation ($r = 0.655$, see Supplementary Fig. 6a).

We used centromeric FISH probes to distinguish local gene copy gains from aneuploidy. Comparing gene counts to centromere counts, we found that *CARD11*, *CDX2*, *EDEM2*, and to a minor extent, *EGFR* displayed aneuploidy (parallel gains in centromere and gene signal counts), whereas *MMP9* showed only local gene amplification in multiple independent samples (Supplementary Fig. 6b). Interestingly, FISH analysis revealed not only significant differences between the specimens, but also showed CNV variability between adjacent cells (usually 50 cells counted per sample) within the same sample (note the large variation in bar graphs Fig. 4b). Taken together our analysis detected different patterns of CNV in CRC, that is, between primary tumours and metastases from the same patient and between neighbouring cells within one tumour (Supplementary Fig. 6b).

**CNVs differ between distinct compartments within a tumour.**
We asked whether genomic heterogeneity also exists between different areas of the same tumour. To analyse intra-tumour heterogeneity, we methodically disassembled an individual stage IIA tumour (Fig. 1c–e). The tumour was divided into 5 blocks, each sectioned and divided into 4 distinct compartments; left and right lateral, and luminal and deep invasive front, yielding 68 independent DNA samples. Each sample thus originates from a spatially distinct, well-defined compartment within the tumour. A separate block containing healthy colon tissue was used as control. We sequenced all DNA samples at a mean coverage of 1,800 reads, using the CRC panel (Fig. 5a, Supplementary Fig. 7 and Supplementary Data 2). We found that all parts of the tumour carried the same *APC* (c.C4099T, p.Q1367*) and *TP53* (c.G524A, p.R175H) variants, and no further SNVs in any individual sample in coding regions covered by our panel (confirmed by Sanger and SNuPE validation; see 'Methods' section).

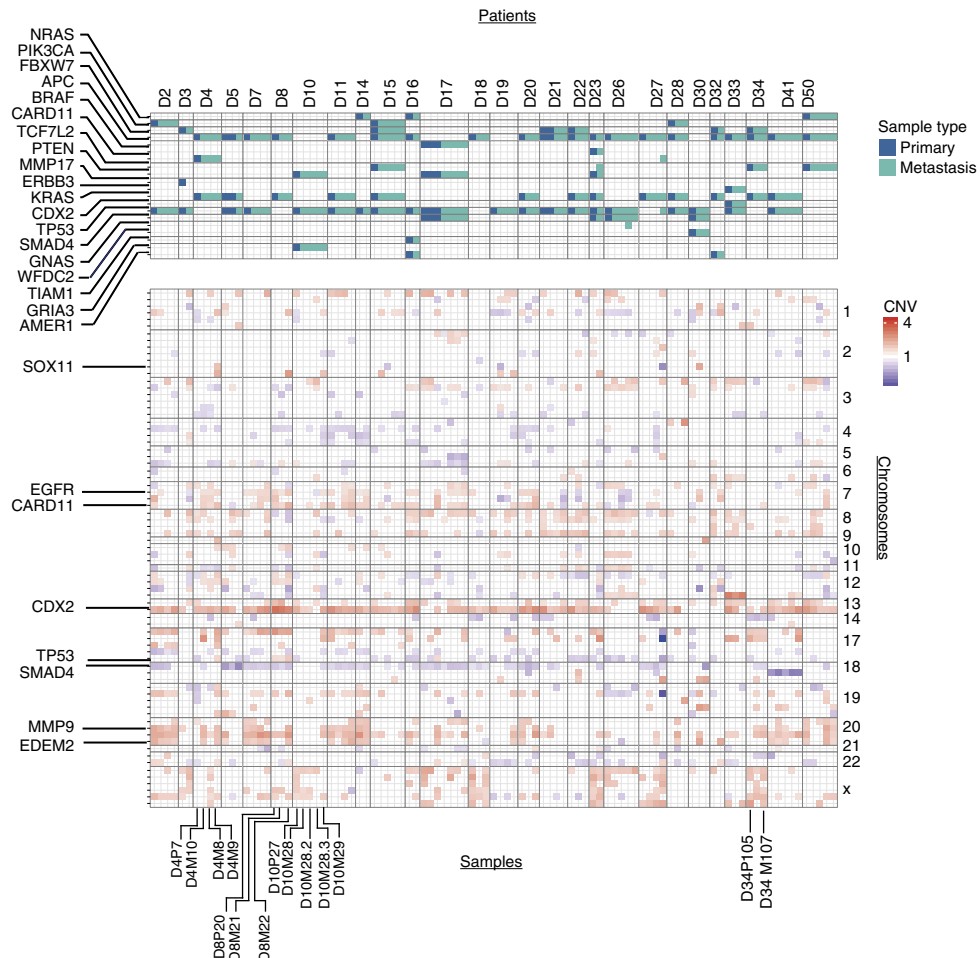

**Figure 3 | Concordant SNVs but discordant CNV profiles in matched primary and metastatic tumours.** In-depth representation of alterations at the SNV level (top) and the CNV level (bottom) for each patient's matched samples. Each SNV is indicated in blue for mutation in primary tumour sample or turquoise for mutation in metastases. CNVs are indicated in red for gain in gene copy number and in blue for loss. Detailed representation in Supplementary Fig. 5.

Our results show that the strong concordance of driver mutations previously detected between primary and metastatic lesions (see above) is also present in different areas of this single tumour. We picked up a few non-coding or synonymous mutations in several regions (Supplementary Fig. 8).

We next investigated CNVs in all 68 samples by CNVPanelizer. We detected typical alterations previously described in CRC, for example, loss of the short arm of chromosome 17 harbouring *TP53* and gain of the long arm of chromosome 20 with *GNAS, EDEM2* and *MMP9* genes[22] (Fig. 5a). Importantly, patterns of CNV differed between the individual samples sourced from different regions of the same tumour.

We independently analysed intra-tumour copy number heterogeneity using FISH analysis on *CARD11, MMP9, BRCA2, EGFR* and *TP53* (12–18 samples per gene; Fig. 5b). Data from FISH confirmed the existence of different levels of CNV within the tumour analysed, and furthermore indicated widespread aneuploidy (Supplementary Fig. 9c). Analysis of CNV by FISH and CNVPanelizer were in good agreement with a correlation coefficient of $r = 0.621$ (Supplementary Fig. 9a). We also used quantitative reverse transcription PCR to validate CNVs affecting *TP53, MMP9, GNAS, EDEM2, SOX9, APC, SMAD4, MAP2K4* and *HDAC3* genes in 43 samples, and once again could confirm the CNVpanelizer results (correlation coefficient $r = 0.7936$, Supplementary Fig. 9b). In addition to FISH and qPCR validation of CNV aberrations, we performed shallow WGS on 10 distinct

areas of this tumour and a matched normal tissue. CNV values as obtained from WGS displayed a strong correlation to those derived from panel sequencing ($r = 0.821$; Fig. 6a). These data corroborate the existence of intra-tumour CNV discordance in CRC.

**3D tumour reconstruction reveals spatial patterns of CNV.** To visualize the spatial distribution of intra-tumour CNV heterogeneity, we assembled a morphological model of this tumour in 3D and superimposed the CNV data onto the reconstructed tumour's architecture, encompassing the different compartments (left and right lateral, luminal and deep invasive front). Mapping of individual CNVs (specifically *MMP9, CARD11* and *BRCA2,* all validated by FISH), onto the 3D model showed regional, but distinct spatial patterns (Fig. 6b–d; see Supplementary Fig. 12 for additional gene patterns and Supplementary Software 1 for an interactive 3D experience). For instance, *BRCA2* and *ATM* were assigned the highest DNA copy number gains in the luminal region, *EDNRB* in the lateral compartments, and *HDAC2* in a subregion of the invasive front.

We performed agglomerative hierarchical clustering of the CNVs recovered in all 68 tumour samples. Two highly divergent clusters were found (Fig. 7a; Supplementary Table 1; and Supplementary Fig. 10). Each cluster was characterized by CNVs in distinct genes: in cluster 1 *CSMD3, PTK2, BRCA2, PREX2* and *BRAF* displayed copy number gains, while *NOTCH3* showed copy

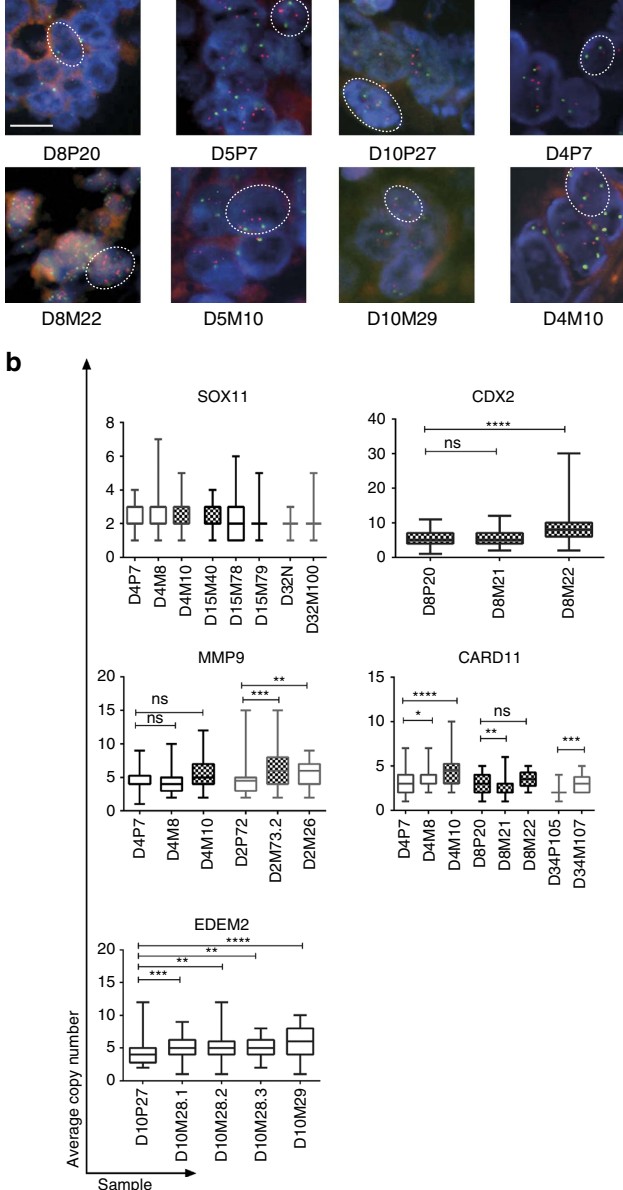

**Figure 4 | FISH analysis of inter-tumour heterogeneity.**
(**a**) Representative FISH for *CDX2, MMP9, EDEM2* and *CARD11* in orange and centromeres of individual chromosomes in green. Scale bar depicts 10 µm. (**b**) Quantification of FISH using probes for *CDX2, MMP9, CARD11* and *EDEM2* reveals differential gene copy numbers between matched primary and metastatic samples. A *SOX11* probe was used as a control. 50 cells randomly selected were counted in each sample. The distribution of signals per sample is depicted. Checkered boxes indicate samples from tumour exposed to chemotherapy. Wilcoxon signed rank test, *$P < 0.05$, **$P < 0.01$ and ***$P < 0.001$. Box shows 25th and 75th percentiles and sample median as horizontal line, whiskers show maximum and minimum points.

number loss. In cluster 2, *CARD11, MMP9* and *HDAC1* were amplified (*P* values in Supplementary Table 2). Strikingly, when the clusters were mapped onto the 3D model, we found that the two clusters were distributed between the proximal and distal axis of the tumour (Fig. 7b, Supplementary Movie 1). Thus, our comprehensive spatial reconstruction revealed CNV differences between and across distinct compartments of a primary CRC, despite stable and uniform patterns of driver mutations.

## Discussion

Our study investigated inter-tumour and intra-tumour heterogeneity in CRC. We found a striking discordance between primary tumours and metastases, as well as within a single tumour at the level of gene copy numbers. Three-dimensional reconstruction of a primary tumour revealed a spatial distribution of chromosomal alterations clustered along the proximal–distal axis of the tumour. In contrast, we found a high concordance in known driver mutations within both the reconstructed tumour and the primary/metastases pairs taken from the same patient. High concordance was present in driver mutations such as *APC, KRAS, NRAS, PIK3CA* and *SMAD4*, in addition to *AMER1, TCF7L2, CARD11* and the EGFR family members *EGFR* and *ERBB3*. Our analysis indicates that gene CNVs are the major source of tumour heterogeneity in CRC development.

We found chromosomal aneuploidy and heterogeneity far more prevalent than hot-spot SNVs within driver oncogenes among multiple tumours between patients and within an individual patient. Importantly, within the limited genomic area the panel covers, some intronic and synonymous mutations show a heterogeneous pattern (Supplementary Figs 4 and 8). This could indicate that genome wide sequencing may additionally reveal clonal SNVs, which our analysis does not cover. CNV heterogeneity was inferred using CNVPanelizer and validated by FISH analyses, ruling out false detection of PCR amplification and FFPE artifacts; the robustness of the CNV results was additionally confirmed by WGS analysis of selected samples. Among the genes amplified was *MMP9*, encoding a matrix metalloprotease, whose overexpression is associated with tumour progression and invasion[23]. Furthermore, *MMP9* copy number gain has been associated with gene overexpression in gastric cancers[24]. Interestingly, we found increased *MMP9* gene copy number in the two metastases of patient D2 when compared with the matched primary sample, as well as intra-tumour heterogeneity of *MMP9* along the proximal–distal axis (Fig. 4b and Supplementary Figs 6b and 9c). Similarly, *CDX2* exhibited copy number gains ranging from duplication to more than 20-fold amplification. CDX2 was identified as a lineage-specific homeobox transcription factor involved in intestinal epithelial cell proliferation and differentiation[25] and has recently been proposed as a prognostic and predictive marker for stage II and III CRC (ref. 26). Our analysis therefore suggests that CNV heterogeneity could endow cells with functional traits important for tumour progression, invasion or metastasis.

The data we presented does not itself lead to functional conclusions on tumour heterogeneity, as would be possible from a joint analysis of CNV and transcriptome data. While other studies have shown clear correlation between CNV and gene expression in gastric cancers[24,27], this need not be the case for all CNV studies. However, due to limited material we were not able to conduct expression studies on our cohort.

Sequencing of multiple neighbouring serial sections from a non-metastasized colon tumour detected ubiquitous mutations in *APC* and *TP53* as well as copy number gains in the *GNAS* and *EDEM2* genes on the long arm of chromosome 20 and consistent loss of heterozygosity of *TP53*. This analysis provides evidence for the existence of a common dominant ancestor clone proposed earlier[28]. However, the analysis also discerned two separate cell populations by CNV analysis and hierarchical clustering. After superimposing the two clusters onto the 3D reconstruction of the tumour, two sub-populations at the proximal and distal ends manifested. This finding supports the so-called 'Big-Bang model' of CRC described recently[4]. The model suggested that a small number of genetic alterations are either public to the entire tumour, or exhibit enrichment at specific sides of the tumour. Our analysis indeed found a high level of DNA copy number

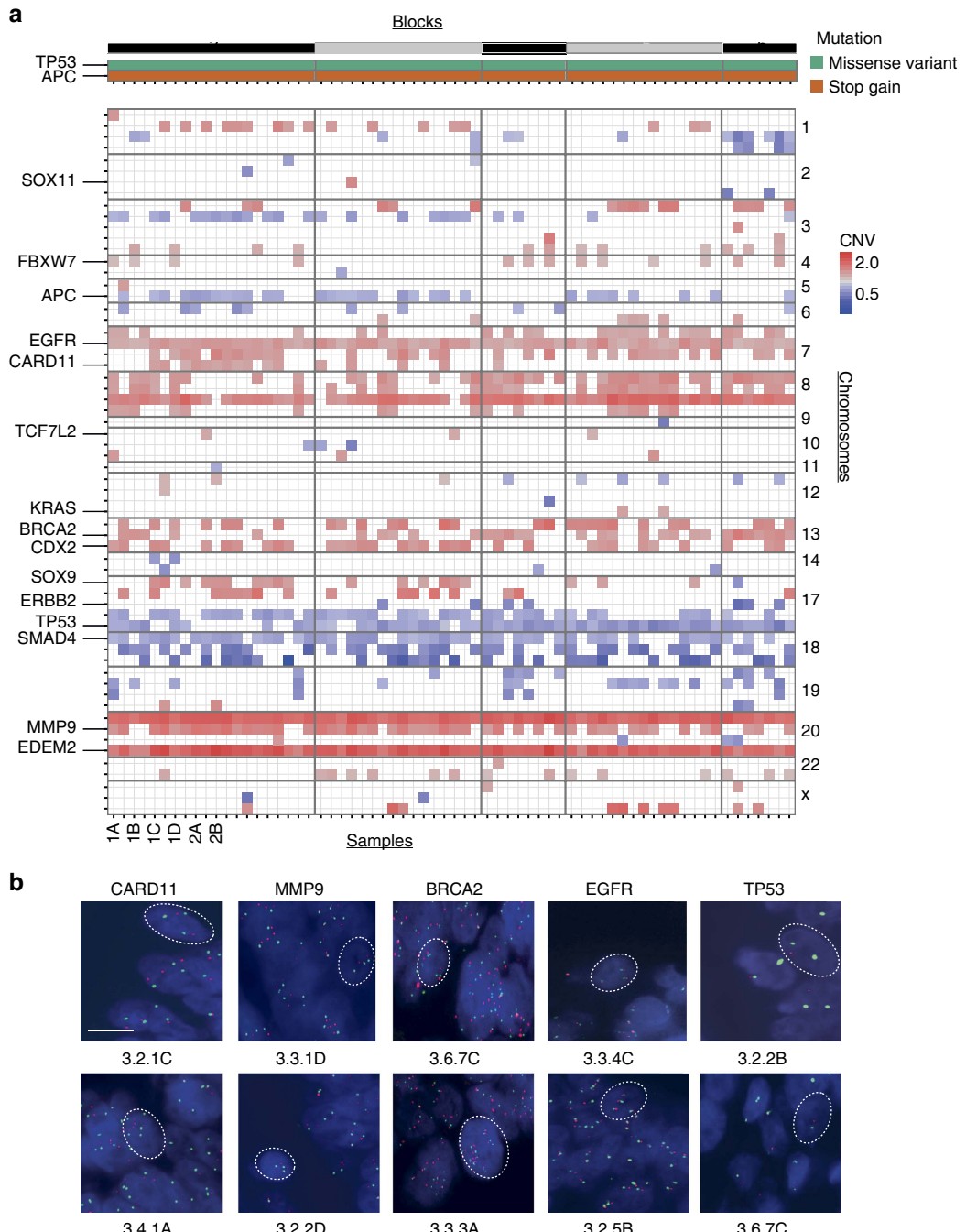

**Figure 5 | Concordant SNV but discordant CNV profiles within one tumour. (a)** Throughout all 68 samples obtained by dissection of a stage II CRC, only a stop-gain *APC* (orange) and a missense *TP53* (green) alteration were detected and validated (top panel). However, genes displaying CNVs were highly heterogeneous throughout the same tumour (lower panel). Copy gain and loss are shown in red and blue, respectively. For a detailed representation, refer to Supplementary Fig. 7. **(b)** Representative FISH for *CARD11, MMP9, BRCA2, EGFR* and *TP53* displayed in orange and for centromeres of chromosomes in green. Twelve to 18 samples were stained per gene, and 30 cells counted per sample. In total, 73 samples were analyzed. Scale bar depicts 10 μm.

heterogeneity between proximal and distal compartments of the tumour. Using FISH, we also discovered DNA copy number heterogeneity at the level of individual cells. However, we cannot distinguish from our data whether this results from a dynamic progressive process or if it is the result of intermingling of different sub-clones during an early 'Big-Bang' scenario. Clearly, our investigation does not reach the sensitivity of single intestinal gland analysis of previous studies[4,29]. However, we distinguish compartments of the tumour–the invasive front, luminal and

lateral areas as well as proximal and distal sides–where we find distinct CNV patterns for multiple genes implicated in tumour development.

Many of the CNVs detected are likely to reflect aneuploidy rather than local gene amplification (Supplementary Figs 6b and 9c). Cancer aneuploidy has been associated with adverse prognosis in several types of cancers[30–32]. It is the most common genomic alteration encompassing >80% of CRCs and has been implicated in drug-resistance[30]. Chromosomal instability is

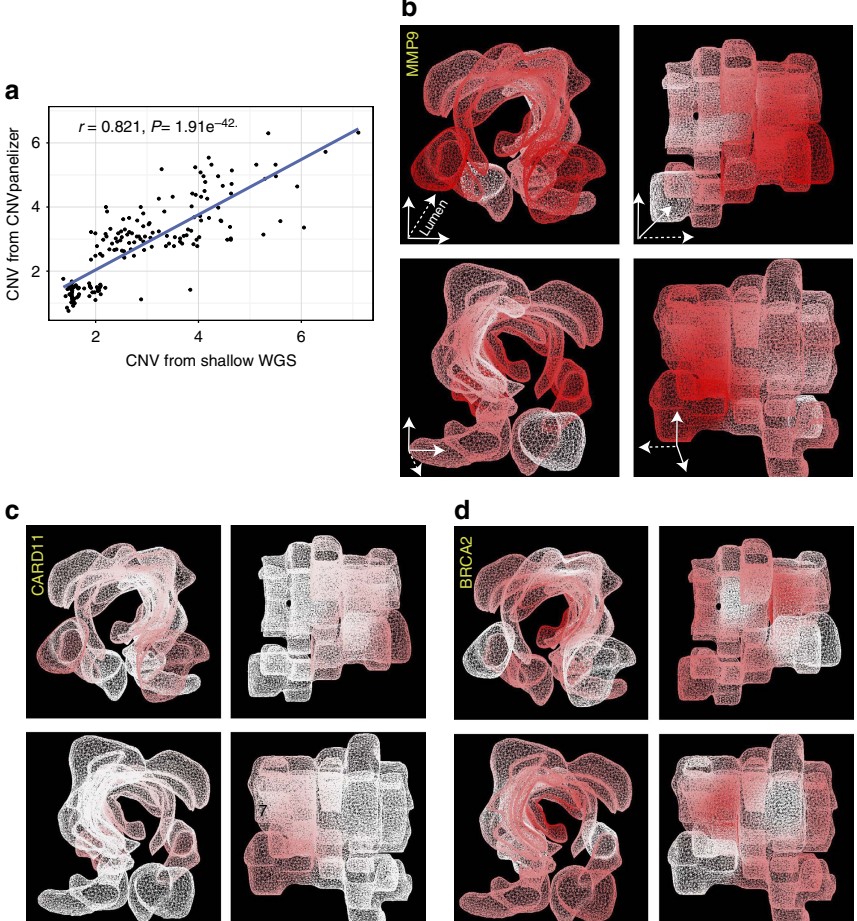

**Figure 6 | 3D architectural reconstruction of a CRC reveals heterogeneous localization of CNVs.** (**a**) Correlation of CNVs recovered by shallow whole-genome sequencing and CNVPanelizer in 10 tumour samples. (**b**) 3D morphological tumour model re assembled with guidance from H&E-stained slides, overlaid with the local distribution of CNVs of *MMP9*, (**c**) *CARD11* and (**d**) *BRCA2* genes; all 3 genes were validated by FISH. Deep and light red colours represent higher, and lower gene copy number gain, respectively. White colour indicates no change in copy number. Insert in **b** for orientation; dotted line indicates the direction to the colon lumen. The tumour image rotates 90° in each quadrant. Both *MMP9* and *CARD11* show strong localization at the proximal–distal axis of the tumour, while *BRCA2* shows less distinct spatial localization.

hypothesized to adopt the role of a 'genomic modifier' once induced via, for example, mitotic stress[33]. This is reflected by our discovery that each tumour sample analysed exhibits an individual CNV pattern, even within a single patient with concordant hot spot SNVs (Fig. 3). Moreover, our 3D model suggests that CNVs might exert this modification potential in spatially different areas within a single tumour. Such a scenario would imply a high potential for tumour adaptation towards targeted and chemotherapy, and would also call for more stringent cancer therapy specifically fatal for aneuploid cells[34].

Previous studies analysing the genetic landscapes of patient-individual primary and metastatic samples in CRC have revealed shared hot-spot mutations[6,16,18]. While our study is in line with these findings, we additionally validated cases of discordance in *MMP17, TCF7L2, GNAS, CARD11* and *TP53*. The *GNAS* mutation affects the G-protein alpha subunit and can potentially impinge on Wnt and MAPK signalling[35,36]. The *MMP17* mutation, localized in close juxtaposition to the peptidase segment of the encoded protein. *MMP17* has been implicated in breast cancer as a positive modifier of EGFR signalling[37]. The *TCF7L2* mutation is located within the beta-Catenin binding domain and potentially modulates Wnt/beta-Catenin signals[38–40]. The *CARD11* and *TP53* mutations detected only in a lung metastasis following chemotherapy could contribute to the activation of the NF-κB pathway[41] and to

abrogation of TP53 function[42,43], respectively. Despite their low prevalence, these discordant mutations can potentially act as drivers of tumour progression or of therapy resistance in rare cases. It is of note that *CARD11* was also highly ranked among the genes displaying intra-tumour CNV. Therefore, mutations and CNV could complement each other in deregulating *CARD11* function.

We propose that future diagnostic approaches should take into account spatial heterogeneity and genomic aberrations on multiple levels, particularly DNA copy number alterations. Our work, encompassing 97 samples from 27 patients plus an extended set of 68 samples from an in-depth analysis of a single tumour has identified candidates for future comprehensive CNV analysis to improve patient stratification.

## Methods

**Patient cohort and sample designation.** Primary colorectal tumours with matched multiple metastasis and normal colon tissue were collected from 27 patients for analysis of inter-patient and inter-tumour heterogeneity. All patients had signed written consent as part of the clinical documentation protocolled of the University Hospital Carl Gustav Carus at the Technical University of Dresden. All tumour specimens were micro-satellite stable CRC. The median age of patients in the cohort was 62 years (range 32–91). Detailed clinical and histopathological data can be found in Fig. 1b and Supplementary Fig. 1.

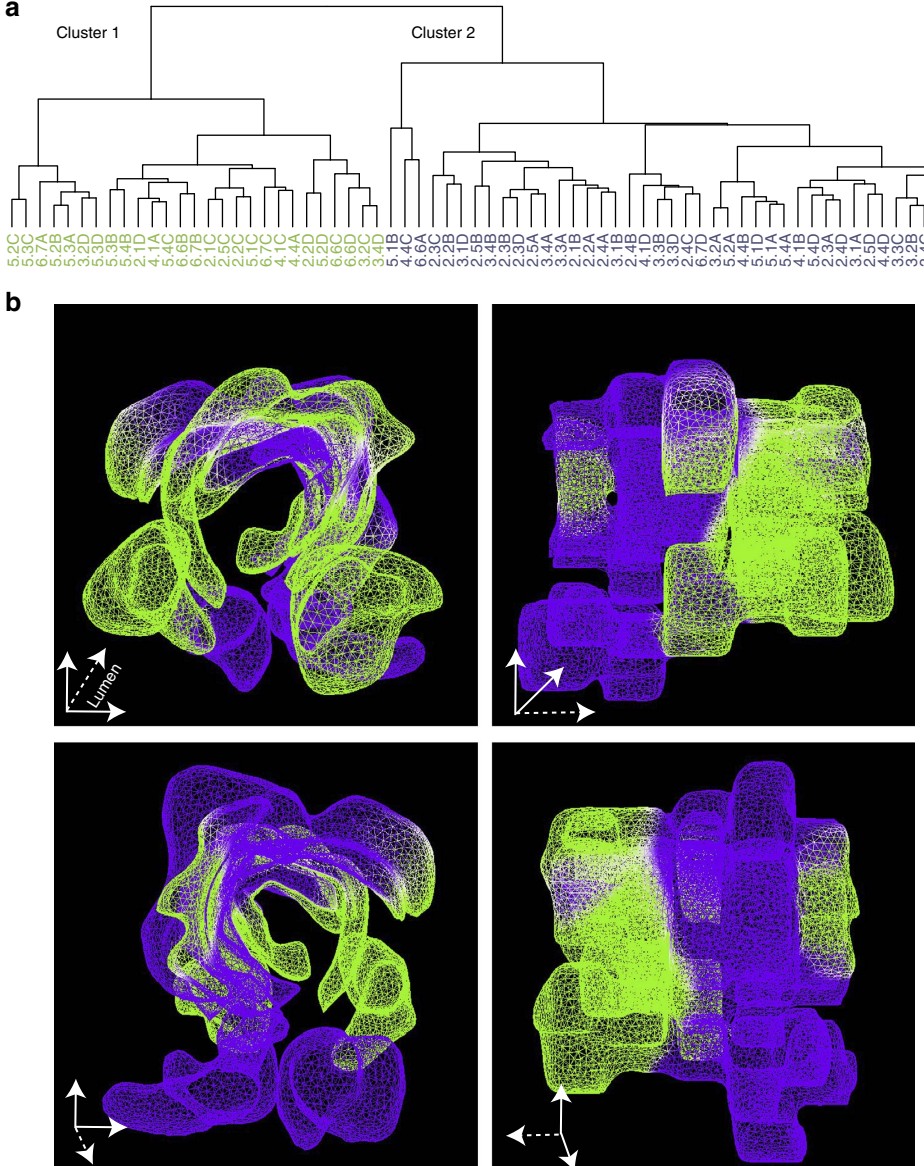

**Figure 7 | Hierarchical clustering reveals distinct localization of CNVs along the proximal–distal axis. (a)** CNVs derived from the 68 samples within the tumour were clustered using agglomerative hierarchical clustering. The objective function used for clustering was the root sum of squared distances between sub-cluster points; the values of each point were the predicted copy number of each gene represented by the panel. Samples clustered into two distinct clusters: cluster 1 and cluster 2. **(b)** To visualize intra-tumour heterogeneity in terms of CNV, a 3D morphological tumour model was generated from slides stained with H&E. The distribution of the two clusters, from **a**, was superimposed onto this 3D model; cluster 1 is depicted in green and cluster 2 in violet. The clusters distinctively localized along the proximal–distal axis. The four images show the 3D model from different orthogonal perspectives.

For intra-tumour heterogeneity analysis, a single-stage IIA CRC tumour was collected at the Charité Universitätsmedizin Berlin, Institute of Pathology, Berlin. The patient was 77 years old at the time of tumour resection. The patient had no lymph node or organ metastasis at time of surgery (TNM pT3pN0(0/17)G2R0L0V0). No neoadjuvant therapy was administered to the patient before tumour isolation. The complete tumour was divided into 5 FFPE blocks, which were subsequently sectioned into 5–10 μm slices. Sections stained with haematoxylin and eosin (H&E) were prepared out of every 15th consecutive section (one group), all other sections were used for DNA isolation and FISH analysis. Before DNA isolation, each section was divided into four parts designated by the letters A, B, C and D corresponding to the left and right lateral tumour regions, and the luminal and deep invasive front compartments, respectively (Fig. 1e), resulting in 68 samples. In addition, healthy colon tissue was isolated and DNA prepared separately. Naming scheme for samples is 'patient' 'block'.'group' plus the letter A–D indicating compartment. Healthy tissue sample has sample I.D.: 3.1-1 (patient 3. block 1.group 1) and no compartment division.

This study has been approved by the ethics commissions at the University Hospital Carl Gustav Carus/Technical University of Dresden (protocol EK59032007) and the Charité Universitätsmedizin Berlin (protocol EA1/260/12).

**Design of the colorectal cancer gene panel.** To investigate the CRC-specific genetic landscape, we designed a custom CRC panel. The panel is comprised of the most commonly mutated or copy-number variant genes in micro-satellite stable CRC (selected from the Cancer Genome Atlas (TCGA)[39] and Genentech's[40] whole-genome sequence databases and the Catalogue of Somatic Mutations in Cancer (COSMIC)[44]), as well as genes known to confer sensitivity or resistance to drugs (as retrieved from the Comparative Toxicogenomics Database (CTD)[45], the Cancer Cell Line Encyclopaedia (CCLE)[46] and the Genomics of Drug Sensitivity in Cancer database (GDSC)[47]). The complete 100-gene panel (Supplementary Fig. 2a,b) included primers for 793 amplicons, covering 125 bp-stretches of mainly exonic regions which were found to be mutated at least in five different specimens present in the databases of TCGA and Genentech.

**Targeted massive parallel sequencing.** Targeted massive parallel sequencing employed Ion Torrent PGM technology (Life Technologies) for primary data generation and a MiSeq device (Illumina) as a second platform. All samples were inspected by a pathologist and tumour tissue was determined by H&E staining. Tumour tissue was macrodissected from slides and DNA was extracted using the

GeneRead DNA FFPE kit (Qiagen). Quality and quantity of DNA was determined by RNAse P quantification (Life Technologies).

Whenever possible, 10 ng of DNA were used for multiplexed PCR amplification with the Ion Ampliseq Library kit (Life Technologies), using two amplicon pools per DNA sample. Samples were ligated to Ion Xpress Barcode Adapters (Life Technologies) and purified using Agencourt AMPure beads (Beckman Coulter). Two sample were combined on a 316v2 chip and sequenced on an Ion Torrent PGM device (Life Technologies) with an average read depth of 1,500 (range 666–3,532) for the inter-tumour heterogeneity cohort and 1,800 (range 920–2,739) for the intra-tumour heterogeneity analysis. Filtered data are available in Supplementary Data 1 and 2 for inter- and intra-tumour studies respectively.

Sequence reads were aligned to the GRCh37 sequence using the Torrent Mapping and Alignment Program (TMAP; Life Technologies). Aligned reads $> 50$ nucleotides with a mapping quality $> 4$ were kept and trimmed to amplicon boundaries, using in-house Python scripts. Variants were called on the processed reads using the Torrent Variant Caller (TVC; Life Technologies) under the 'strict' setting as specified by the IonTorrent Suite. A further filter was added to remove variants within homopolymer regions $> 4$ nucleotides in length and variants that were also found in the normal tissue yielding reliable somatic variants. All variants from samples belonging to the same patient were then merged. Reference and alternate read counts were extracted directly from the original read alignments for the samples in which the matching variants were not called using an in-house Python script.

Each variant was annotated with several types of biological information using an annotation framework called SoFIA[48]. A threshold of quality $Q = 50$ was applied to reduce false positive results due to FFPE material (Supplementary Fig. 2c).

**MiSeq (Illumina) re-sequencing.** We used a MiSeq device (Illumina) as a second platform for validating sequencing results for patients D11 and D30 using the same custom CRC panel. DNA (10 ng) from FFPE embedded tumour tissue was prepared using the Ion AmpliSeq Library Kit 2.0 (Life technologies) for the CRC panel, followed by library preparation using the NEB Next Ultra DNA Library Prep kit (end repair, A-tailing, adaptor ligation and amplification; NEB, E7370S) and NEB Next Multiplex Oligos provided for Illumina (NEB, E7335S). 100 pM of resulting library DNA was pooled, all samples from one patient were pooled together. The average read output was $4.6 \times 10^6$ reads.

**Shallow whole-genome sequencing.** DNA from samples 3-4D, 5-3D, 6-6B, 5-2D, 5-4B, 2-2B, 2-3B, 2-4A, 3-3A and 3-4A was prepared with TruSeq Nano DNA Library Prep Kit (Illumina) according to the manufacturer's protocol: 100 ng of genomic DNA were fragmented to 350 bp using Covaris LE220 system (Covaris, Inc.). Fragments were end-repaired, A-tailed, adaptor ligated and PCR amplified (8 cycles). The final libraries were validated using Agilent Tapestation 2200 (Agilent Technologies) and Qubit flourometer (Invitrogen), normalized and pooled in equimolar ratios. 101 bp paired-end sequencing was performed on the Illumina HiSeq 4000 according to the manufacturer's protocol. To match downstream analysis requirements, only the first members of the read pairs were considered, and were truncated to the first 50 bp. The resulting genome coverage was between 0.25 and 0.39 × (0.64–0.79 × for paired end reads). Data analysis were performed following Scheinin et al.[49]

**Validation of single-nucleotide variations.** To validate NGS results we used the Illumina platform, Sanger sequencing and Single Nucleotide Primer Extension/ HPLC method (SNuPE). MiSeq (Illumina) platform was used to re-sequence several samples (described above). Sanger sequencing was used to validate SNVs that occurred at an allele frequency higher than 10%, either using primers from our CRC panel or newly designed primers. For mutations with allele frequencies below 10% we used SNuPE, as described in Hoogendoorn et al. and Tierling et al.[50,51] This method allows validation of variants with allele frequencies down to 1%. (see Supplementary Tables 3,4 and 6 for primer lists and Supplementary Fig. 3 for validation examples).

**Prediction of DNA copy number variations from NGS data.** DNA CNV were predicted by analysing the panel-sequencing data using CNVPanelizer[21], which compares tumour samples with a pool of non-matched normal tissue samples. The algorithm combines bootstrapping the reference set with the subsampling of amplicons associated with each of the target genes. This serves as a non-parametric distribution estimation of the gene-wise mean ratio between healthy reference samples and each tumour sample. To correct for different numbers of amplicons per gene, a second subsampling step was applied. For the inter-patient and tumour heterogeneity study we used the matched normal tissue as a reference. For the intra-tumour heterogeneity study, blood samples from healthy individuals and matched normal tissue were used as the reference. PCR duplicates were removed and only CNVs with a $P$ value $< 0.001$ in both the bootstrapping and background methods were called.

**Fluorescence in situ hybridization.** FISH was performed on 5 μm tumour sections. We used commercially available and standardized probes for detection of EGFR (Vysis LSI EGFR SO/CEP7 SG), TP53 (Vysis LSI TP53; Abott Molecular), SOX11, CDX2, CARD11, BRCA2, EDEM2 and MMP9 (Empire Genomics). Hybridization was performed according to manufacturer's instructions. Where possible we scored 50 cells per sample (inter-patient and inter-tumour heterogeneity study) and 30 cells per sample (intra-tumour heterogeneity study) for hybridization patterns using an Olympus microscope. Analysis was conducted using 'BioView solo' (Abbott Molecular).

**Quantitative polymerase chain reaction analysis.** Four tumour samples and matched normal tissue were analysed using 0.5 ng of FFPE DNA. An internal control was integrated using normal colon tissue DNA from two patients without a history of cancer. Analysis was performed using the $2^{(-\Delta\Delta Ct)}$ approach. qPCR was performed for APC, EDEM2, GNAS, HDAC3, MAP2K4, MMP9, TP53, SMAD4 and SOX9 genes, the TERT gene served as a reference (Supplementary Table 5 for a list of primers).

**Study of genetic and patient characteristics.** The relationships between the genetic attributes of the tumours in the cohort and several patient characteristics were investigated through the application of a series of tests to reveal potential targets and biomarkers, and to find mechanistic explanations for observed genetic variations. Two genetic attributes (mutation status and gene copy number) were tested against (1) whether the tumour was a primary neoplasm or metastatic, (2) whether the patient had received treatment or not, (3) disease stage, (4) whether the primary tumour was in the right or left colon, (5) gender of the patient, (6) whether the samples was derived from a lymph-node metastasis, (7) whether the sample was derived from a liver metastasis. Primary tumours (P) and metastases (M) were analysed together for characteristics 1 and 2. Primary tumours and metastases were analysed separately for characteristics 3–5. Only metastases were analysed for characteristics 6 and 7.

We corrected for inter-patient genetic similarity (the patient effect) using a generalized linear mixed model with the 'logit' link function to model the binomial nature of the phenotype values. In cases where there was only one sample from each patient, a generalized linear model with a 'logit' link function was fit. In all other cases, both the independent variable and the response variable were binomial in nature and a Fisher Exact test was used.

The $P$ values for the generalized linear models and generalized linear mixed models were calculated using a chi-square test comparing the fitted model with the null model in which only the patient effect was modelled. For each independent/ response variable combination the $P$ values were corrected for multiple testing using the Benjamini–Hochberg method. All analyses were implemented using R and the libraries nlme (ref. 52) and ggplot2 (ref. 53).

**Hierarchical clustering of genes.** CNVs belonging to the 68 samples from the intra-tumour heterogeneity study were clustered using pvclust (ref. 54), an agglomerative hierarchical clustering algorithm that performs multiscale bootstrap resampling to calculate an approximate unbiased $P$ value for each identified cluster. Bootstrapping was performed with 10,000 repetitions. The genes that best distinguish between the clusters were determined by applying the non-parametric Wilcoxon rank-sum test to the difference in gene copy number aberrations between the clusters for each gene individually. The resulting $P$ values were corrected for multiple testing using the Benjamini–Hochberg method. The genes with the most significant $P$ values were chosen for validation according to the methods described above.

**Reconstruction of tissue architecture and 3D visualization.** To obtain a full spatial representation of the tumour and visualize intra-tumour genetic heterogeneity, we generated a 3D micro-anatomic tumour model from consecutive sections taken from all 68 tumour samples from one stage II CRC patient. H&E stained slides were digitized with a 3D Panoramic 250 scanner (3D HISTECH). Four different compartments (designated A, B, C and D, see Fig. 1e) were annotated manually for each digital slide corresponding to the region used for separate molecular analyses[55]. The shapes of these annotations were exported and scaled into images of $1,776 \times 562$ pixels. As slides from five different blocks were used to create the 3D model, a manual registration of those images was necessary. Slices in the $x$-$y$-plane were stretched along the $z$-axis to preserve uniform aspect ratio. As a consequence of the tumour's overall macroscopic architecture, the number of sections differs among tissue blocks. For this patient, block #1 represents normal tissue and was not used for reconstruction, block #2 is represented by 20 samples, block #3 by 16, block #4 by 8, block #5 by 16 and block #6 by 8 samples. Slight deformations of the sections are unavoidable due to the softness of the tube-like colonic tissue. When serial tissue sections were placed into embedding cassettes, discontinuities between them were minimized by careful processing, however not eliminated completely. The position and rotation of regions was adjusted manually, while the shapes were not changed (rigid registration). In a fine adjustment step each region under-went further minimal non-rigid transformation to improve reconstruction. Next, the MoMo (morphology modeller) algorithm described in Klauschen et al.[56] was used to

create a 3D model for each region of the planar mask images containing the shape information. Models were stored in.obj files and imported into Blender (https://www.blender.org/), a 3D modelling tool, which was used for additional smoothing of the model and exporting it into one.fbx file. Colouring information was loaded from.csv files and passed to a shader that interpolates a value for each vertex from all measured values depending on the squared distance between the center of the measured region and the vertex. The data from the shader is mapped back to the tissue sections by assigning the respective molecular feature value of a subregion to the center voxel of the subregion in each histological plane for which the molecular analysis was performed. Then the surface mesh colours are interpolated between neighbouring centres. We used the 3D reconstruction and colouring process to model the spatial distribution of the clusters found through hierarchical clustering and the copy number for the genes MMP9, CARD11 and BRCA2 in addition to CNV cluster information obtained by the method described above. An interactive 3D visualization of the measured data (per gene and clusters) can be downloaded by the reader using Supplementary Software 1. Supplementary Table 7 lists the measured CNV data used for the 3D visualization shown in the screenshot Fig. 6b–d. Representative macro images of tissue H&E sections used for 3D reconstruction can be found in Supplementary Fig. 11.

**Data availability.** Sequence data has been deposited at the European Genome-phenome Archive (EGA), which is hosted by the EBI and the CRG, under accession number EGAS00001002150. The remaining data are available in the article or the Supplementary Files or from the authors on request.

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

## Acknowledgements

We thank Stefanie Mende for FISH analysis and Tabea Trampert for PCR analysis. We thank 'High throughput Sequencing' of the Genomics & Proteomics Core Facility, DKFZ, for providing excellent Whole-Genome Sequencing services. We thank Jeroen van Marle for Supplementary Movie preparation. The authors gratefully acknowledge funding by German Ministry of Education and Research (e:Bio OncoPath 0316184A to C.S., R.S., U.L. and M.M.; DKTK to C.S., R.S., W.W. and D.A.) and the Deutsche Forschungsgemeinschaft (BSIO to C.S., U.L. and B.G.).

## Author contributions

C.S. conceived the project. C.S., S.M. and F.K. conceived experiments. S.M., K.M., A.M., F.M., S.Z., B.G., D.L., S.T. and M. Moebs performed experiments. L.H.C., S.M., D.H., E.B., F.K. and P.D. analysed data. C.S., S.M., L.H.C. R.S. and M. Morkel wrote the manuscript. D.A., G.F. and F.K. provided clinical samples. C.O. and T.W. developed the CNVpanelizer. U.L., R.S., D.B., W.W., H.B., D.S. and T.R. provided scientific and technical support. M.v.W. provided pathological assessment of all slides.

## Additional information

**Competing financial interests:** The authors declare no competing financial interests.

