## [Peer Review File · Nature Communications]

Reviewers' comments:

Reviewer #1 (Remarks to the Author): Expert in 3D image reconstruction

This manuscript explores the inter-patient and intra-patient/tumour heterogeneity of colorectal cancers. I have expertise of the 3D reconstruction aspects of this work and confine my remarks to that.

The authors describe how they have taken a single tumour sample, cut it into 5 distinct blocks along the proximal-distal direction then processed each block to produce serial sections from a paraffin embedded block. Reported section thickness was 5-10 microns and every 15th section was stained with H&E for imaging and delineation of 4 specified tumour regions. Within these sections each region was manually drawn and taken into a system to reconstruct surfaces from each region set after manual alignment. The subsequent surfaces were smoothed and rendered using Blender. Colour coding of the vertices allowed the visualisation of the patterns of specific gene and CN variations throughout the sample taking data from the intervening 14 sections not used for the reconstruction. Overall this 3D reconstruction is very simplistic and provides a limited representation of the spatial patterns in the data. This process is described but there are a number of additional details that should be provided:

1/ what is the section thickness actually used, does the repositioning of regions in 3D space reflect the variable sampling thickness? Is there uniform sampling of the space or some blocks have more sections, others less?

2/ Is each boundary of each region aligned independently of other regions from the same section? The text suggests that this is the case but then the relative positions of these regions would be changed and the 3D visualisation of all subregions in one view invalid.

3/ How is the data mapped back into the space? Does the spatial density of say a CNV become mapped in the x-y plane from the observation in another section or is the data summarized per tumour region (left, right, luminal, deep invasive) then equal values assigned to the corresponding boundaries in the model?

4/The visualisations of figure 6 are very "blocky" presumably reflecting the 5 blocks and the low sampling frequency with each block. Viewed from the side there is clearly very little continuity between the proximal-distal block which should in principle align. Why is this the case? Is it tissue processing and embedding artefacts post slicing of the sample or an alignment issue?

The visualisations in figure 6 are difficult to decode. If the colouring for each gene is based on the two dimensions of distal-proximal axis then the tumour-region category then this data will be much better presented as a simple 2D table with the regions on one axis and the distal-proximal position on the other. This would be much easier to read and compare patterns. If however the 3D models hold real x-y patterning information then this would be better represented as a volumetric data-set rather than colours on the surfaces of the regions. The use of the 3D visualisation needs to be further justified.

It would be very useful for the reader to provide representative images of the H&E stained sections with the tumour regions used depicted for each of the blocks to allow a better understanding of the definition of the terms used and allow the reader to judge the variability of the manual interpretation. In addition the data-files used for visualisation should be made available to allow users to read the data into Blender or other package for deeper exploration of the patterns. The manuscript can only provide a pre-selected 2D view which does not do justice to the data.

In addition a visualisation of the tumour regions i.e. its "anatomy" would significantly help the interpretation of the renderings provided in figure 6.

Reviewer #2 (Remarks to the Author): Expert in colorectal cancer genetics

Massively parallel sequencing technology or so called next-generation sequencing have been revealing intra-tumor and inter-tumor heterogeneity of somatic alterations in cancer patients. The authors investigated coding mutations and copy-number alterations in 27 patients with colorectal cancer (CRC), and showed that copy-number alterations rather than coding mutations are prominent as the inter-tumor heterogeneity between primary and metastatic tumors. The authors further carried out three dimensional (3D) analyses on the genetic alterations in a single case of primary CRC to demonstrate the existence of two clusters of copy-number alterations along the proximal-distal axis within the tumor. This is a cutting-edge manuscript that is informative and valuable for diverse groups of oncologists and genome researchers.

The following are minor issues to be addressed:

(1) Title. The authors carried out heterogeneity analyses on 27 cases with CRC and detailed 3D analysis on a single case of primary tumor. Because the authors did not carried out 3D analysis in 27 cases, the title of this manuscript might be relatively exaggerated. The authors need to refine the title of this manuscript.

(2) Terminology of NGS. 'NGS' is frequently used for massively parallel sequencing technology; however, I am afraid that massively parallel sequencing technology is current generation sequencing technology rather than next generation. Therefore, the authors are advised not to use 'NGS' without definition in the title and running title.

(3) CRC panel and deep sequencing. The authors need to describe the originality of their CRC gene panel and deepness of their targeted sequencing in the Abstract.

(4) CRC gene panel. The authors originally chose 100 genes frequently altered in CRC, and listed them in supplementary Figure 2A. To promote usefulness for the readers, the authors are advised to add a supplementary illustration panel of human chromosomes 1 ~ 23, X and Y on which their hundred genes are plotted.

(5) Lack of expression data. Because amplified genes are not always expressed, lack of expression data is a weak point in this manuscript. The authors need to discuss this issue as a weak point in the Discussion section.

Reviewer #3 (Remarks to the Author): Expert in tumour heterogeneity and genomics

The authors used gene panel sequencing to analyze genomic alterations in colorectal tumors in order to address the topic of genetic tumor heterogeneity. To this aim, they also analyzed 27 cases, where primary tumors and metastases of the same patient could be compared. They also analyzed sub-segments of one individual tumor in great detail. While SNVs appeared to be by enlarge stable between the different tumors of one patient, DNA copy number alterations displayed considerable variations. While it might be intuitively difficult to match those findings, such data would certainly have great influence on our current thinking of tumor heterogeneity and tumor evolution.

Therefore, it is very important to consider the robustness of the CNV scoring as performed in this study. CNV scoring is very robust when using whole genome sequencing data, do-able but less robust when using whole exome sequencing data, and do-able with an even smaller part of the

genome - like small panels of a few genomic segments - however with a robustness that needs to be assessed. With only 100 small segments of the genome analyzed in this study, the robustness of CNV scoring needs to be convincingly demonstrated.

Such scoring could be, e.g., greatly influenced by variable percentages of tumor cells (versus normal cells) in the analyzed samples. Thus, independent scoring of CNVs would be required. Whole genome sequencing (1x coverage would be sufficient for high quality CNV scoring) would be the method of choice. The authors offer a comparison of CNV scores produced by FISH of a very limited set of data points.

However, as a matter of fact, correlations between FISH and CNV scores are basically not existent. R2 scores of 0.30 and 0.43 do not strongly support an association (as also illustrated by the clouds of data points in the respective figures), and there seems to be the misunderstanding that p-values would make a meaningless R2 value meaningful.

It seems a bit awkward that the authors address the existence of potential subclones always to „spatially separated" areas. So far it is not clear, whether such subclones are always spatially distinct and the authors do not provide information that would support this concept. Subclones might as well be intermingled and might simply not be detected because they amount only to a very small percentage of the cell population that is molecularly analyzed.

The manuscript is put together in a very sloppy manner. Some examples are:

- While at other places of the manuscript the term is correctly used, the term „SNV" is wrongly introduced as „somatic nucleotide variations".
- CNV is always referred to as „copy number variations" without mentioning a single time that „DNA copy number variations" are meant.
- The first sentence of the third paragraph of the results section refers to „inter-patient heterogeneity" while inter-tumor heterogeneity is meant.
- and so on at many (!) places of the text
- The exemplary FISH images with arbitrary outlines of DAPI stained nuclei often do not even match with the obvious areas of an individual nucleus

Minor comment:

The term „ploidy" is used when entire chromosomes or the whole chromosome set are considered, but not for the description of individual gene copies. Thus, the term „CARD11 ploidy" is wrong.

Point-by-point-reply to the reviewers' comments.

Reviewer #1 (Remarks to the Author): Expert in 3D image construction

This manuscript explores the inter-patient and intra-patient/tumour heterogeneity of colorectal cancers. I have expertise of the 3D reconstruction aspects of this work and confine my remarks to that.

The authors describe how they have taken a single tumour sample, cut it into 5 distinct blocks along the proximal-distal direction then processed each block to produce serial sections from a paraffin embedded block. Reported section thickness was 5-10 microns and every 15th section was stained with H&E for imaging and delineation of 4 specified tumor regions. Within these sections each region was manually drawn and taken into a system to reconstruct surfaces from each region set after manual alignment. The subsequent surfaces were smoothed and rendered using Blender. Color coding of the vertices allowed the visualization of the patterns of specific gene and CN variations throughout the sample taking data from the intervening 14 sections not used for the reconstruction. Overall this 3D reconstruction is very simplistic and provides a limited representation of the spatial patterns in the data. This process is described but there are a number of additional details that should be provided:

REPLY: We agree that our reconstruction method is not highly sophisticated (though we would prefer the term “simple” over “simplistic”). However, the purpose of our 3D visualization is not an extremely precise reconstruction of the true colon, as it would be necessary for several applications. Instead, our point is to show the highly biased distribution of a certain property of the samples, i.e., their amount of specific CNVs. However, to make this distribution now easier to grasp for the reader, we added new information. Supplementary figure 11, Supplementary table 1 and additional file 1 now cover comments related to the 3D image construction in the methods section.

1/ what is the section thickness actually used, does the repositioning of regions in 3D space reflect the variable sampling thickness? Is there uniform sampling of the space or some blocks have more sections, others less?

REPLY: The tumor was cut into 5 parts and each of them was positioned in a separate paraffin block. From these, sections were prepared until the tissue material was almost completely used up. Section thickness was 10µm for DNA isolation/sequencing, 5µm for HE-staining and FISH analysis. Sampling was performed largely uniformly, apart from the technical variability of manual tissue sectioning. This was done by an experienced technician used to prepare hundreds of tissue sections per week. Because sections were used for FISH and DNA analysis, these sections were not available for 3D morphological reconstruction and therefore, slices in the x-y-plane had to be stretched uniformly along the z-axis to guarantee a realistic aspect ratio. As a consequence of the tumor's overall macroscopic architecture, the number of sections differs among tissue blocks. For example, block #2 is represented by 20 samples, block #3 by 16, block #4 by 8, block #5 by 16 and block #6 by 8 samples. Block #1 represents the normal control specimen.

To better explain this process, we modified the methods section ‘3D visualization of intra-tumor heterogeneity and reconstruction of tissue architecture’

2/ Is each boundary of each region aligned independently of other regions from the same section? The text suggests that this is the case but then the relative positions of these regions would be changed and the 3D visualization of all subregions in one view invalid.

REPLY: The regions were indeed not aligned independently. Instead, we performed non-rigid transformation of the regions for improved alignment and reduction of the tissue processing/embedding artifacts as described below (see REPLY 4). We have modified the methods sections '3D visualization of intra-tumor heterogeneity and reconstruction of tissue architecture' to *"The position and rotation of regions was adjusted manually, while the shapes were not changed (rigid registration). In a fine adjustment step each region underwent further minimal non-rigid transformation to improve reconstruction."*

3/ How is the data mapped back into the space? Does the spatial density of say a CNV become mapped in the x-y plane from the observation in another section or is the data summarized per tumour region (left, right, luminal, deep invasive) then equal values assigned to the corresponding boundaries in the model?

REPLY: The data from the shader is mapped back to the tissue sections by assigning the respective molecular feature value of a subregion to the center voxel of the subregion in each histological plane for which the molecular analysis was performed. Then the surface mesh colors are interpolated between neighboring centers.

To clarify this issue, we modified the manuscript in the methods section: 3D visualization of intra-tumor heterogeneity and reconstruction of tissue architecture.

4/The visualizations of figure 6 are very "blocky" presumably reflecting the 5 blocks and the low sampling frequency with each block. Viewed from the side there is clearly very little continuity between the proximal-distal block which should in principle align. Why is this the case? Is it tissue processing and embedding artefacts post slicing of the sample or an alignment issue?

REPLY: The reviewer is right in assuming that the blocky appearance of the reconstructed tumor is due to artifacts caused by tissue processing and paraffin embedding. Slight deformations of the sections are unavoidable due to the softness of the tube-like colonic tissue. When serial tissue sections were placed into embedding cassettes, discontinuities between them can be minimized by very careful processing, however not eliminated completely.

We modified the manuscript in the methods section: 3D visualization of intra-tumor heterogeneity and reconstruction of tissue architecture

5/The visualizations in figure 6 are difficult to decode. If the coloring for each gene is based on the two dimensions of distal-proximal axis the tumor-region category then this data will be much better presented as a simple 2D table with the regions on one axis and the distal-proximal position on the other. This would be much easier to read and compare patterns. If however the 3D models holds real x-y patterning information then this would be better represented as a volumetric data-set rather than colors on the surfaces of the regions. The use of the 3D visualization needs to be further justified.

REPLY: We have obtained gene sequence information for each of the tumor's sub-regions, the left, right, luminal and deep invasive compartments, respectively. Each of the data points has distinct x-y-z coordinates representing one of the above regions in each tissue slice. To overlay this information onto the "tissue architecture" of the tumor, we find a mesh representation to be superior to a volume-rendering approach, which would not be easily compatible with the anatomic surface mesh. Since each sub-region is represented by serial sections, we obtained three-dimensional information with regard to the entire tumor architecture. We believe that a 3D model provides a much more intuitive visualization of the data than with a table. However, we agree that tables also give important information which is not deducible from an image. Therefore, we followed the reviewer's suggestion and added 'Supplementary table 1' containing the numeric values.

7/ It would be very useful for the reader to provide representative images of the H&E stained sections with the tumor regions used depicted for each of the blocks to allow a better understanding of the definition of the terms used and allow the reader to judge the variability of the manual interpretation. In addition the data-files used for visualization should be made available to allow users to read the data into Blender or other package for deeper exploration of the patterns. The manuscript can only provide a pre-selected 2D view which does not do justice to the data. In addition a visualization of the tumor regions i.e. its "anatomy" would significantly help the interpretation of the renderings provided in figure 6.

REPLY: We added a representative HE histological image of each block: Supplementary Fig. 11. We think that giving a reader a chance to create its own views is an excellent suggestion. Unfortunately, blender files do not hold coloring information as color is added in the last rendering step. In order to nevertheless allow the reader to explore the model and the coloring information, we added the files necessary for running the interactive 3D visualization as 'Additional file 1'. In addition we have created a GIF movie file which nicely exhibits the rotating 3D tumor image with the CNV cluster information overlaid onto it, 'Supplementary video 1'.

Reviewer #2 (Remarks to the Author): Expert in colorectal cancer genetics

Massively parallel sequencing technology or so called next-generation sequencing have been revealing intra-tumor and inter-tumor heterogeneity of somatic alterations in cancer patients. The authors investigated coding mutations and copy-number alterations in 27 patients with colorectal cancer (CRC), and showed that copy-number alterations rather than coding mutations are prominent as the inter-tumor heterogeneity between primary and metastatic tumors. The authors further carried out three dimensional (3D) analyses on the genetic alterations in a single case of primary CRC to demonstrate the existence of two clusters of copy-number alterations along the proximal-distal axis within the tumor. This is a cutting-edge manuscript that is informative and valuable for diverse groups of oncologists and genome researchers.

The following are minor issues to be addressed:

(1)Title. The authors carried out heterogeneity analyses on 27 cases with CRC and detailed 3D analysis on a single case of primary tumor. Because the authors did not carry out 3D analysis in

27 cases, the title of this manuscript might be relatively exaggerated. The authors need to refine the title of this manuscript.

REPLY: We thank the reviewer for this comment. We now changed the title to:

DNA copy number changes define spatial patterns of heterogeneity in colorectal cancer

(2) Terminology of NGS. 'NGS' is frequently used for massively parallel sequencing technology; however, I am afraid that massively parallel sequencing technology is current generation sequencing technology rather than next generation. Therefore, the authors are advised not to use 'NGS' without definition in the title and running title.

REPLY: Thank you for the observation. We have substituted the term NGS by massive parallel sequencing, where appropriate.

(3) CRC panel and deep sequencing. The authors need to describe the originality of their CRC gene panel and deepness of their targeted sequencing in the Abstract.

REPLY: Very good suggestion. We have added this information in a brief form to the abstract and double-checked that sequencing depth is properly explained within the methods section and in the results.

(4) CRC gene panel. The authors originally chose 100 genes frequently altered in CRC, and listed them in supplementary Figure 2A. To promote usefulness for the readers, the authors are advised to add a supplementary illustration panel of human chromosomes 1 ~ 23, X and Y on which their hundred genes are plotted.

REPLY: We agree. We added such a figure as Supplementary Fig. 2 (see image below)

(5) Lack of expression data. Because amplified genes are not always expressed, lack of expression data is a weak point in this manuscript. The authors need to discuss this issue as a weak point in the Discussion section.

REPLY: We are aware that the lack of expression data is a shortcoming of our study. The reason is a lack of material. Providing sections for RNA isolation and analysis would have reduced material for DNA sequencing and further reduced the resolution of the reconstructed tumor that was criticized by reviewer #1. Following the reviewer's critique, we now address this issue in the discussion section on page 10.

Reviewer #3 (Remarks to the Author): Expert in tumour heterogeneity and genomics

The authors used gene panel sequencing to analyze genomic alterations in colorectal tumors in order to address the topic of genetic tumor heterogeneity. To this aim, they also analyzed 27 cases, where primary tumors and metastases of the same patient could be compared. They also analyzed sub-segments of one individual tumor in great detail. While SNVs appeared to be by enlarge stable between the different tumors of one patient, DNA copy number alterations displayed considerable variations. While it might be intuitively difficult to match those findings, such data would certainly have great influence on our current thinking of tumor heterogeneity and tumor evolution.

Therefore, it is very important to consider the robustness of the CNV scoring as performed in this study. CNV scoring is very robust when using whole genome sequencing data, doable but less robust when using whole exome sequencing data, and do-able with an even smaller part of the genome - like small panels of a few genomic segments - however with a robustness that needs to be assessed. With only 100 small segments of the genome analyzed in this study, the robustness of CNV scoring needs to be convincingly demonstrated. Such scoring could be, e.g., greatly influenced by variable percentages of tumor cells (versus normal cells) in the analyzed samples. Thus, independent scoring of CNVs would be required. Whole genome sequencing (1x coverage would be sufficient for high quality CNV scoring) would be the method of choice.

REPLY 1: As a response to this reviewer's comment we performed shallow (low depth) whole genome sequencing (WGS) on 10 tumor samples and 1 distantly located normal healthy tissue and estimated CNVs using Scheinin et al (*DNA copy number analysis of fresh and formalin-fixed specimens by shallow whole-genome sequencing with identification and exclusion of problematic regions in the genome assembly. Genome Res. 24, 2022–2032 (2014)*). Comparing the results with the original CNVpanelizer output derived from panel sequencing revealed a high correlation, which further increased our trust in the original method. These new data have been added to Figure 6 (figure below) and accompanying text.

The authors offer a comparison of CNV scores produced by FISH of a very limited set of data points. However, as a matter of fact, correlations between FISH and CNV scores are basically not existent. R² scores of 0.30 and 0.43 do not strongly support an association (as also illustrated by the clouds of data points in the respective figures), and there seems to be the misunderstanding that p-values would make a meaningless R² value meaningful.

REPLY: We conducted FISH analysis to support our findings on genomic heterogeneity derived from CNV analysis of the panel data and already validated by qPCR. We think that the concordance of FISH and panel/CNVpanelizer result is quite good, as can be seen from Supplementary figures 6a and 9a in the new manuscript (Figures 4c and 5c in the old manuscript). Clearly, correlations between results from a FISH analysis and a sequencing-based analysis cannot be expected to be perfect, as FISH is a single-cell technique while sequencing results are obtained from bulk tissue sections. Besides, our new data using shallow WGS fully support our previous findings.

Nevertheless, we agree that usage of R² correlation was misleading. We now replaced this measure with Pearson correlation which does not assume linear dependencies. In addition, in the updated graphs we now only take into account results from the CNVpanelizer which have passed the threshold indicating significance, please refer to Oliveria et al. (page 6 of documentation <https://www.bioconductor.org/packages/3.3/bioc/html/CNVPanelizer.html>) Furthermore, we followed your suggestion and removed p-values from all correlation measures.

It seems a bit awkward that the authors address the existence of potential subclones always to „spatially separated" areas. So far it is not clear, whether such subclones are always spatially distinct and the authors do not provide information that would support this concept. Subclones might as well be intermingled and might simply not be detected because they amount only to a very small percentage of the cell population that is molecularly analyzed.

REPLY 2: We concur that the term ‘spatial subclones’ is misleading and therefore we removed it from the manuscript. However, to describe geographically distinct areas within the tumor we continue to use the term ‘spatially separated area’. For example in the abstract where we describe

macroscopically distinct areas routinely defined in tumor pathology such as the invasive front or a near-lumen region. In addition the new version of our manuscript now places more emphasis on the different spatially separated compartments; left and right lateral fronts and the luminal and invasive regions.

The manuscript is put together in a very sloppy manner. Some examples are:

- While at other places of the manuscript the term is correctly used, the term „SNV“ is wrongly introduced as „somatic nucleotide variations“.

REPLY 3: Sorry. We corrected this error.

- CNV is always referred to as „copy number variations“ without mentioning a single time that „DNA copy number variations“ are meant.

REPLY 4: We actually did explicitly refer to DNA copy number variation in the abstract (in the original submission). But we agree that the main body should also use the full term and now added the term “DNA” to CNV throughout the text.

- The first sentence of the third paragraph of the results section refers to „inter-patient heterogeneity“ while inter-tumor heterogeneity is meant.

REPLY 5: We apologize for this error which we corrected.

- and so on at many (!) places of the text

REPLY 6: We carefully went through the entire paper multiple times to make sure that such problems no longer occur.

- The exemplary FISH images with arbitrary outlines of DAPI stained nuclei often do not even match with the obvious areas of an individual nucleus

REPLY 7: We have improved the stringency of the outlines on FISH images including the one related to patient tumor 8 and the CDX2 gene which we assume is the image the reviewer is referring to. The slides were re-considered, we have improved nucleus detection and pictures were taken from the routine archival material used for our study to the best of our ability. We believe that our approach is sufficiently robust in general and relevant for many studies based on archival tissue in translational oncology. Such tissues, particularly when collected in multi-centric trials, often exhibit fixation artifacts and variations due to long-term storage.

Minor comment:

The term „ploidy" is used when entire chromosomes or the whole chromosome set are considered, but not for the description of individual gene copies. Thus, the term „CARD11 ploidy" is wrong.

REPLY8: We have corrected this.

REVIEWERS' COMMENTS:

Reviewer #1 (Remarks to the Author):

I have re-read the manuscript in conjunction with the authors' remarks and find the text much improved in terms of the presentation and description of the 3D reconstruction. The addition of a movie-clip in the supplementary data helps and the view of the histology sections and how they are marked up provides a clear view of the approach. It is clear that the techniques will deliver some measure of the 3D distribution of the CNVs but it is certainly "simple" and a crude "first-pass". The additional data makes this clear.

I have no further comments with respect to the 3D reconstruction.

Reviewer #2 (Remarks to the Author):

In this revised submission, the authors appropriately addressed the issues that I pointed out in their initial submission.

Reviewer #3 (Remarks to the Author):

The authors have adequately addressed all points of critique that I had raised

Point-by-point response to any issues raised by our referees.

REVIEWERS' COMMENTS:

Reviewer #1 (Remarks to the Author):

I have re-read the manuscript in conjunction with the authors' remarks and find the text much improved in terms of the presentation and description of the 3D reconstruction. The addition of a movie-clip in the supplementary data helps and the view of the histology sections and how they are marked up provides a clear view of the approach. It is clear that the techniques will deliver some measure of the 3D distribution of the CNVs but it is certainly "simple" and a crude "first-pass". The additional data makes this clear.

I have no further comments with respect to the 3D reconstruction.

We thank the reviewer for her / his comments.

Reviewer #2 (Remarks to the Author):

In this revised submission, the authors appropriately addressed the issues that I pointed out in their initial submission.

We thank the reviewer for her / his comments.

Reviewer #3 (Remarks to the Author):

The authors have adequately addressed all points of critique that I had raised

We thank the reviewer for her / his comments.